# Emergent topological polarization textures in relaxor ferroelectrics

Maksim Eremenko [1,2,3] ✉, Victor Krayzman[1,3], Semen Gorfman[4], Alexei Bosak[5], Helen Y. Playford [6], Philip A. Chater [7], Bruce Ravel[1], William J. Laws[1], Feng Ye [2], Arianna Minelli [2], Bi-Xia Wang[8], Zuo-Guang Ye [8], Matthew G. Tucker [2] & Igor Levin [1] ✉

Relaxor ferroelectrics underpin high-performance actuators and sensors, yet the nature of polar heterogeneities driving their broadband dielectric response remains debated. Using a unified, multimodal structural refinement framework— simultaneously fitting complementary X-ray and neutron total scattering, X-ray absorption spectra, and diffuse scattering—we reconstruct 3D mesoscale polarization maps in the classic relaxor system $PbMg_{1/3}Nb_{2/3}O_3–PbTiO_3$. We uncover self-organized swirling polarization textures with half-skyrmion (meron) vortices, challenging models of independent polar nanoregions. These textures, characterized by smooth changes in the polarization direction, originate from overlapping volumes in which the projections of locally correlated polarization vectors onto each volume's long axis share the same sign. Vortex cores correlate strongly with local charge and strain gradients imposed by compositional heterogeneities. In this work, our results suggest that chemical disorder, acting via depolarizing and strain fields, stabilizes topological vortex textures of the polarization field, offering a route for engineering new dielectric and ferroelectric functionalities.

Local perturbations of atomic order caused by competing chemical bonding interactions or chemical heterogeneities are increasingly acknowledged to impact the functional properties of various solid-state materials. Relaxor ferroelectrics, or relaxors, are regarded as the epitome of this phenomenon. Relaxors are distinguished by a broad frequency-dependent maximum of dielectric permittivity, often linked to chemical disorder[1]. These materials enable commercially viable formulations with exceptional electromechanical and energy-storage properties[2–4]. Extensive research has been conducted to relate such properties to the underlying chemistry and structure, producing a general theoretical framework and empirical relationships currently guiding the materials' development[5]. The relaxor behavior is widely accepted to be caused by local variations in chemistry, which create polar inhomogeneities perceived as defining the dielectric response. However, a seven-decade quest to understand or at least adequately describe these inhomogeneities has been unsuccessful, with this task remaining a fundamental challenge[6].

A canonical example of such systems is the perovskite-like relaxor $PbMg_{1/3}Nb_{2/3}O_3$ (PMN) and its solid solutions with $PbTiO_3$ (PT)[2], which demonstrate outstanding electromechanical performance. This performance of PMN-PT, especially in the single-crystal form, is attractive for medical imaging, sonar, and other sensor and actuator applications. Numerous attempts to establish the relationship between the structure and properties in PMN-PT suggested several, sometimes seemingly contradictory, scenarios, but no ultimate clarity[6]. The proposed depictions of the polarization vary from isolated polar

---

[1]Materials Measurement Science Division, National Institute of Standards and Technology, Gaithersburg, MD, USA. [2]Spallation Neutron Source, Oak Ridge National Laboratory, Oak Ridge, TN, USA. [3]Theiss Research, La Jolla, CA, USA. [4]Department of Materials Science and Engineering, Tel Aviv University, Tel Aviv, Israel. [5]European Synchrotron Radiation Facility, Grenoble, Cedex, France. [6]ISIS Facility, Science and Technology Facilities Council, Didcot, Oxfordshire, UK. [7]Diamond Light Source, Science and Technology Facilities Council, Didcot, Oxfordshire, UK. [8]Department of Chemistry and 4D LABS, Simon Fraser University, Burnaby, BC, Canada. ✉e-mail: eremenkom@ornl.gov; igor.levin@nist.gov

nanoregions (PNRs) embedded in a non-polar matrix[7] to polar nanodomains separated by domain walls[8] to PNRs separated by wide boundaries featuring gradual changes in polarization[9–11]. Still, some works questioned even the very concept of PNRs and their significance in explaining the relaxor properties[12]. A principal reason for this ambiguity is the inherent multilevel complexity of the relaxor structures, which combine chemical disorder with dynamic and static polar displacements that evolve upon changing temperature[13,14]. Yet, each measurement method can capture only certain aspects of this complexity, making it difficult to reconstruct a three-dimensional (3D) polarization vector field and its relationship to the underlying chemistry over all the relevant length and time scales.

It is well established that PMN maintains an average cubic structure even at very low temperatures. In this state, Mg and Nb, octahedrally coordinated by oxygen, undergo partial rock-salt-type ordering on the nanoscale. Recent results[10,15] have confirmed earlier inferences from dark-field transmission electron microscopy (TEM)[16] of the nanoscale regions with stronger ordering separated by those featuring a gradient of the order parameter. The partial disorder of Mg and Nb frustrates the off-center shifts of Pb atoms, which are stabilized by the hybridization of the Pb 6$s$ and O 2$p$ electrons and contribute significantly to the polarization. Refinements of the PMN structure using large-scale atomic configurations revealed a hierarchical assembly of PNRs featuring cooperative cation displacements relative to the oxygen framework[10]. The neighboring small-scale nanoregions with the polarization aligned parallel to the ⟨111⟩ directions were predominantly ≈71° variants. The agglomerates of several of these nanoregions exhibiting the head-to-tail arrangement of their polarization vectors appeared to form larger-scale PNRs.

The essential characteristics of structural snapshots from these refinements[10] agreed with molecular dynamics (MD) simulations[9]. Both sets of results indicated a lack of a non-polar matrix, instead revealing PNRs separated by regions with a gradually changing direction and magnitude of polarization akin to wide, low-angle domain walls. Atomic-resolution scanning TEM images (STEM)[11] supported a multi-domain state with a high density of such domain walls, with their locations pinned by structural heterogeneities, such as regions of stronger chemical ordering or areas with larger octahedral distortions and tilting. The large-box structural refinements[10] and MD simulations[9] agree on a multi-domain state responsible for the observed anisotropic X-ray and neutron diffuse scattering (DS). Still, the specific correlation topology contributing to this anisotropy remained unclear. The proposed hierarchical assembly of PNRs separated by wide, low-angle boundaries hints at the existence of inter-PNR correlations. However, such correlations have not been considered, even though they could be essential for clarifying the mechanisms of excellent electromechanical properties in PMN-PT crystals.

Here, we addressed this question by explicitly recovering interatomic correlations over the ten-nanometer length scale in PMN and PMN-PT compositions (with 30 % and 35 % PT) in their high-temperature cubic state. Our approach involved refining large-scale structural models simultaneously against multiple types of experimental data, including neutron and X-ray total scattering, extended X-ray absorption fine structure, and 3D reconstructions of X-ray DS. Advancements in data-analysis techniques and refinement software achieved as a part of this work permitted a significantly deeper understanding of the relaxor structure than was previously achievable.

In this work, we provide evidence for three primary effects not covered by the existing models. First, the nanoscale chemical ordering is enhanced in regions with a higher content of larger, weakly polarizable Mg ions. These regions disrupt longer-range correlations among polar displacements and alter the polarization vector field. Second, cation displacement correlations are multi-level, with correlation volumes that extend across multiple PNRs. Third, the inter-PNR correlations yield 3D maze-like textures of continuously curling

polarization that contain vortex-like formations, such as merons, acquiring net dipole moments. The development of these topological textures is attributed to the mitigation of the depolarizing electric fields that arise from the nanoscale compositional and charge heterogeneities.

## Results

### Local ordering vs compositional fluctuations

We focused on the cubic ergodic state below the so-called Burns temperature[17], which signifies the onset of extended correlations among polar displacements. This choice allows for a straightforward comparison of different PMN-PT compositions, emphasizing the crystal-chemical trends that are precursors to the lower-temperature (non-ergodic) states. Additionally, it helps to avoid the refinement complexities in cases of monoclinic and tetragonal twinning encountered in PMN-PT at lower temperatures[18]; an effective methodology for such refinements is still under development. Details of refinements and the fitting results are discussed in the Methods section and SI (Fig. S1–S13).

In PMN, the short-range rocksalt-type cation order (Figs. 1a and S14) is similar to previous reports[10]. The refined order parameter and correlation length differ slightly from the published values due to our more accurate treatment of the diffuse intensity scale and width. For PMN-PT compositions with three B-site species, a description of ordering requires three independent short-range order parameters[19]. The results are summarized in Tables S1 and S2 and Fig. S15. The rocksalt-type ordering for Nb and Mg is retained, but its degree and spatial extent are significantly reduced compared to PMN.

In PMN, the local order parameter (see Methods) for Mg and Nb is closely linked to the local Mg content, peaking for a 1:1 ratio of the two B-site species (Fig. 1b). That is, the regions with stronger ordering are enriched with Mg, leading to the enhanced and more extended nanoscale fluctuations in lattice strain and electric charge (Fig. 1c) compared to a random distribution of the B-site cations [see End Notes]. As demonstrated in prior works[10,14], the magnitude of Pb displacements in PMN increases approximately linearly as the number of Mg atoms, $n$, in the [PbMg$_n$Nb$_{8-n}$] clusters increases; this trend is driven by the bonding requirements of the oxygen atoms. The same behavior occurs in the PMN-PT compositions. For a given local Mg/Nb ratio, the displacements are more significant for a larger number of Ti around Pb. While both the local order parameter and the magnitude of the Pb displacements scale with the local Mg concentration, the direct statistical correlation between the two is weak.

The oxygen framework accommodated the differences in the ionic radii of Mg$^{2+}$ (0.72 Å), Nb$^{5+}$ (0.64 Å), and Ti$^{4+}$ (0.605 Å)[20], with the [MgO$_6$] octahedra expectedly expanding and the [NbO$_6$] (weak change) and [TiO$_6$] octahedra contracting relative to the average (Fig. S16). The magnitude of this deformation scales with the local chemical order parameter: stronger ordering allows for larger oxygen shifts along the B-O-B' bonds. The PMN structure at 300 K contained small clusters (1 nm to 2 nm) with the in-phase $a^0a^0c^+$-type[21] octahedral tilting (Fig. 1D) and a rotation angle of ≈4°. At 490 K, the degree of such rotational ordering was reduced, and the coherency length of these tilts was limited to the nearest-neighbor octahedra. No rotational ordering was observed for the PMN-PT compositions.

### Correlated polar displacements: correlation length vs. PNR size

In PMN at 300 K and 490 K and in PMN-PT at 533 K, the Pb ions are offset from their average cubic position (Figs. S17–S19). However, for the PMN-PT configurations, the magnitude of the Pb off-center shifts is reduced compared to PMN (Figs. S19), in line with the shrinkage of the average lattice (Figs. S1). The preferred directions of Pb displacements also change from ⟨111⟩ and ⟨100⟩ for PMN to ⟨100⟩ for PMN-PT. (Figs. S20). Regardless of composition, the displacements of Nb favor the ⟨111⟩ (primary) and ⟨110⟩ directions. In contrast, the Ti cations in PMN-PT are preferentially shifted along ⟨100⟩ (Figs. S21 and S22).

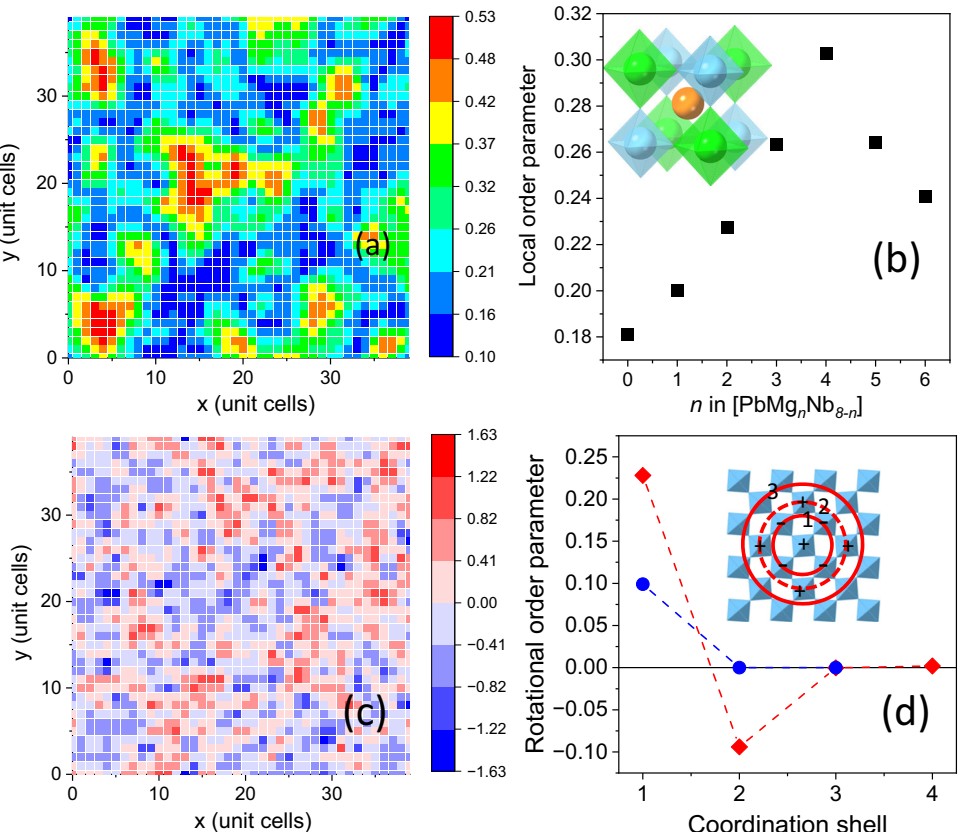

**Fig. 1 | Nanoscale chemical ordering and compositional segregation.** Results from the analysis of the PMN configurations refined using the experimental data shown in Figs. S3, S4, S6 and S7. **a** Single layer of Pb with the color scale reflecting the average local B-site (Mg, Nb) order parameter in the [PbB$_8$] coordination clusters centered on the A-site Pb cations in this layer. This local order parameter, $\eta$, was calculated for each B-cation (see Methods) and then averaged over the eight B cations surrounding each Pb ion. **b** Relationship between $\eta$ and the number of Mg cations in the [PbMg$_n$Nb$_{8-n}$] coordination clusters. Inset shows an example cluster with $n = 4$ and complete rocksalt ordering of Mg (green) and Nb (blue). **c** Same layer of Pb as in **a** but with the color scale reflecting the net formal electric charge (in $e$) for each Pb-centered unit cell. This charge varies with the local Mg$^{2+}$/Nb$^{5+}$ ratio, and the map reveals the extended, positively (Nb$^{5+}$-rich) and negatively (Mg$^{2+}$-rich) charged regions. **d** Order parameter, $\kappa(\iota)$ (see Methods), for octahedral rotations in PMN within layers of [BO$_6$] octahedra around the axis normal to the layer plane. (red) 300 K; (blue) 490 K. The inset illustrates a perfectly ordered pattern of rotations maintaining the octahedral rigidity. Signs of rotations are indicated using "+" and "−" symbols. Circles label successive coordination shells $\iota = 1,2,3$ for the octahedron in the center.

The local alignment of the cation displacements can be characterized using the average angle, $\alpha$, between the displacement vector of a given cation and those of its nearest cation neighbors[10] (e.g., coordination environments [PbPb$_6$], [PbB$_8$], or [BPb$_8$]) (Figs. S23). The angles $\alpha \approx 0°$ and $\alpha \approx 180°$ indicate parallel and anti-parallel alignments. An atomic configuration with randomly oriented displacements yields $\langle\alpha\rangle \approx 90°$ (here, the angular brackets denote the configuration average).

For [PbPb$_6$] in PMN, we obtained $\langle\alpha\rangle \approx 77°$ and $\langle\alpha\rangle \approx 82°$ at 300 K and 490 K, respectively, indicating a preference for the parallel alignment (from our experience and testing, any deviations from $\langle\alpha\rangle = 90°$ greater than 5° are deemed significant). Likewise, for the PMN-PT configurations, $\langle\alpha\rangle \approx 80°$. Dense clusters of atoms with small values of $\alpha$ (e.g., $\alpha < 45°$) can be regarded as PNRs[10]. In PMN, at 300 K, for the [MgPb$_8$] clusters, $\langle\alpha\rangle \approx 88°$, whereas for [NbPb$_8$], $\langle\alpha\rangle \approx 73°$ – consistent with a weakly polarizable nature of Mg$^{2+}$ ions, which are much less effective in supporting displacement correlations than smaller, higher-charge species, like Nb$^{5+}$. For the PMN and PMN-PT configurations, the $\alpha$-values for the [PbPb$_6$] clusters scale positively with those for the [PbB$_8$] units centered on the same Pb atoms (Fig. S24), suggesting that better alignment of displacements between Pb and its neighboring B-cations (distance=$a\sqrt{3}/2$, where $a$ is the lattice parameter) enhances the alignment within the Pb-Pb coordination sphere (distance = $a$).

We are interested in spatial correlations among atomic displacements, which, as refined, are obscured by randomness associated with thermal disorder and the nature of the refinement procedure (e.g., Fig. 2a). We extracted correlated components of displacements using the Fourier filtering of the calculated DS amplitude, with regions of interest selected by placing spherical masks around all Bragg reflections (see Methods). This filtering highlights regions featuring the patterns of atomic displacements or chemical ordering, giving rise to specific diffuse features in diffraction space. Indeed, after applying the filter, the correlations are seen much more clearly (Fig. 2b). In PMN at 300 K, the filtered components for Pb yield $\langle\alpha\rangle \approx 21°$, indicative of a much stronger local alignment compared to the total displacements with $\langle\alpha\rangle \approx 77°$. Those Pb atoms that contribute the most to the DS, as reflected in their acquisition of the largest intensity in the Fourier transform of the DS amplitude, have Mg-deficient local environments with a lesser degree of the B-cation ordering. That is, higher local concentrations of Mg weaken the correlations among the Pb displacements, consistent with Mg acting as a "blocking" species[22].

We found that the array of correlated Pb displacements can be represented as a 3D assemblage of overlapping regions, which, if approximated as ellipsoids, exhibit one principal axis significantly longer than the other two. Within each ellipsoid, the Pb displacements projected onto this long axis point in the same direction. The overlap

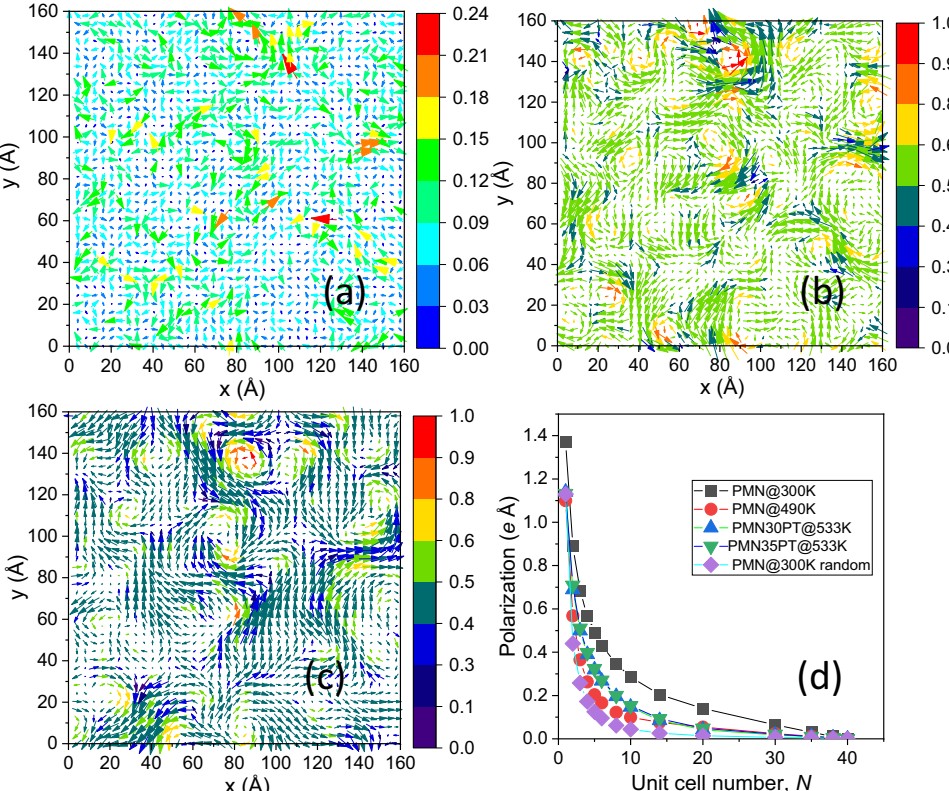

**Fig. 2 | Revealing polar correlations in PMN.** Projections of the Pb displacements onto the (001) plane in the refined PMN (300 K) configuration, averaged over the atomic columns, with vectors indicating the Pb column displacements. Color – displacement magnitude in Å. **b** Same as **a** but for the Pb displacements derived via the Fourier filtering of the calculated diffuse scattering amplitude (see Methods for the description of this filtering). Color—the $Q$-criterion describing the vorticity of the 2D vector field (see Methods). **c** Projection map similar to **a**, **b** but with the vector field corresponding to the local polarization calculated from the refined coordinates of all the atoms in the configuration and formal ion charges (see Methods). Color—the Q-metric as in (**b**). The patterns in **b**, **c** differ somewhat but are strongly correlated. **d** Dependence of the local polarization on the averaging volume of $N \times N \times N$ unit cells for PMN at 300 K and 490 K and for 0.7PMN-0.3PT and 0.65PMN-0.35PT at 533 K. The dependence for a spatially random distribution of the local polarization in PMN at 300 K is shown as a reference (here, the magnitudes and directions of the polarization vectors are the same as in the refined configuration, but any spatial correlations are scrambled). The polarization map shown in (**c**) corresponds to $N = 2$.

of such volumes creates a 3D maze of smaller PNRs. The displacements within the PNRs that belong to several volumes represent a vector sum of the corresponding components.

To visualize these textures, we consider projections of the refined Pb displacements onto the {001} plane in Figs. 2a, b; similar-type vector fields for displacements of atomic columns, albeit purely static, would be revealed in atomic-resolution (S)TEM images. The DS (Figs. 3a, b) computed from such projections reveals the same anisotropy as seen in our 3D datasets (Figs. S4 and S5). We then decompose the column displacements in Fig. 2b into their orthogonal ±[11] (Fig. 3c) and ±[1−1] (Fig. 3d) components; these directions were selected based on the anisotropy of the diffuse scattering. Each component forms stripe-like clusters of positive and negative displacement directions, yielding bicontinuous textures with extended correlation lengths and faceting along the respective ±[11] or ±[1̄1] axes. Overlying both components (Fig. 3e) produces patchy domains in the total projection (Fig. 3f), with displacements in the overlapping regions being a vector sum of the components. That is, the displacements in the adjacent PNRs are correlated.

Simulations based on these 2D patterns show that to obtain streak-like DS around the *HK (H, K ≠ 0)* spots, the displacements must be aligned with the long axis of the correlated regions; otherwise, this intensity would appear cross-like as seen around the *H*0 and 0*K* peaks. The same principle holds in 3D, where displacement components maintain their correlation across multiple PNRs, with the correlation extending longest along the direction of these shifts.

The **q**-dependence (vector **q** defines the deviation from a Bragg position) of the DS around the Bragg peaks in PMN at 490 K differs from that at 300 K; however, the characteristic "tails" transverse to **Q** = ⟨110⟩* remain similarly extended at both temperatures (Fig. S25). The substitution of PT into PMN results in more isotropic and narrower distributions of the diffuse intensity (Fig. S25), indicating longer real-space correlation lengths in PMN-PT, which is consistent with prior works[23]. Our refined models reproduce these changes. However, the correlation lengths for the local polarization (Fig. 2d) calculated directly from the atomic coordinates (see Methods), actually decrease with increasing temperature (for PMN) and PT content. This apparent contradiction arises because the correlations that define the distribution of the diffuse intensity span several PNRs (Fig. 3). The greater extent of these correlations in PMN-PT can be attributed to a smaller concentration of the blocking Mg species and weaker cation ordering, leading to a weaker Mg segregation. The correlation length of polarization in PMN decreases markedly from 300 K to 490 K. The correlation length in both PMN-PT compositions at 533 K is greater than that in PMN at 490 K; however, it is still substantially smaller than this length in PMN at 300 K.

Previous studies invoked ellipsoidal PNRs in a matrix to explain the anisotropy of the diffuse intensity, with both oblate and prolate shapes proposed[24,25]. However, these models fail to predict the correct **q**-dependence of the diffuse intensity[26]. Our findings reveal a principally different perspective, where such volumes with correlated displacements exist only when considering specific

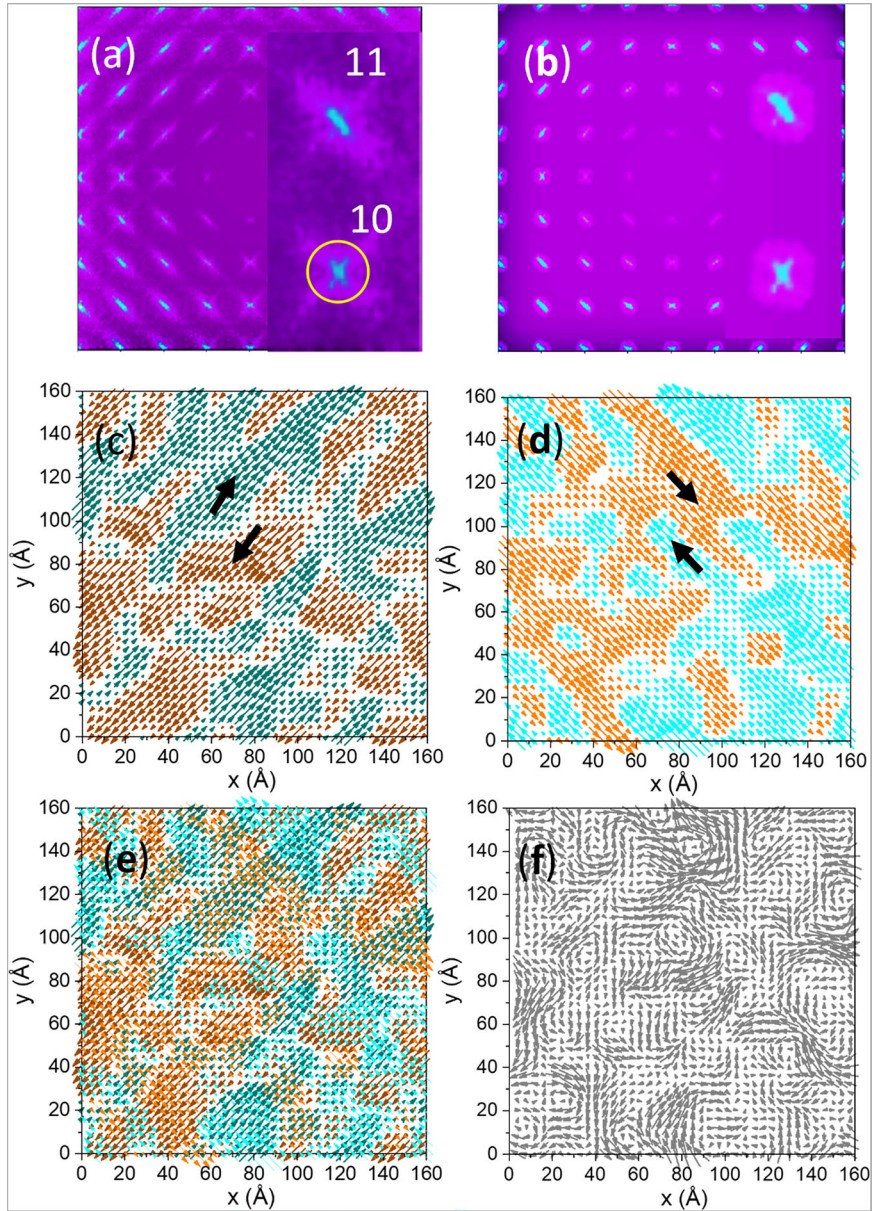

**Fig. 3 | Inter-PNR correlations and origins of diffuse scattering. a, b** Diffuse intensity calculated for the projection maps in Fig. 2a, b, respectively. In both panels, insets show a magnified view of the diffuse intensity around the [10] and [11] Bragg peaks (indices are in 2D). The circle in the inset in **a** illustrates the size of the mask used in the Fourier filtering to yield the displacements shown in Fig. 2b. **c, d** [11] and [1−1] components, respectively, of the displacements in Fig. 2b. In each panel, the two colors highlight the positive and negative displacement directions. **e** Overlay of (**c, d**). **f** Grayscale rendering of Fig. 2b included here to facilitate the side-by-side comparison with (**e**). The vector sum of the two components in **e** can be seen reproducing the total displacement in (**f**).

displacement components rather than the total displacements. The shapes of these volumes, which overlap, yielding intertwined PNRs, explain the overall anisotropy of the diffuse scattering but not the complete intensity distribution that includes contributions from inter-PNR correlations. The size of these "elementary" PNRs, also referred to as α-PNRs in ref. 10, which are characterized by having a distinct direction of the total polarization, can be smaller (2 to 3 times in projection, Fig. 3) than the correlation length for specific displacement components. The low-angle α-PNR boundaries imaged in TEM[11,18] are a product of such inter-PNR correlations.

The divergence calculated for a vector field representing the correlated Pb displacements (See Methods) varied with the local B-site chemistry, from preferentially positive for the Nb-rich environments to negative for the Mg-rich ones (Fig. S26). A similar trend was obtained for the local polarization. This indicates that Nb-rich and Mg-rich

regions act as sources and sinks for polarization, respectively, consistent with other experimental findings and simulations[27]. Spatial distributions of the divergence metric for PMN at 300 K and 490 K were correlated, suggesting this effect to be at least partly static, also in line with it being captured by STEM images[27]. In the PMN-PT compositions, both Mg- and Ti-rich clusters exhibited a preferentially negative divergence of the Pb displacements.

The displacements of Nb are aligned with those of Pb (Fig. S27), resulting in a relatively narrow Nb-Pb distance distribution. In contrast, the displacements of Pb and Mg are only weakly correlated, which, combined with the enhanced magnitudes of Pb displacements around Mg, yields a broad and distorted distribution for the Mg-Pb distances. Interestingly, Ti displacements are *anti-correlated* with those of the neighboring Pb atoms (Figs. S27 and S28), as in the soft mode in $PbTiO_3$[28]. This anti-correlation causes Ti to be off-centered within the

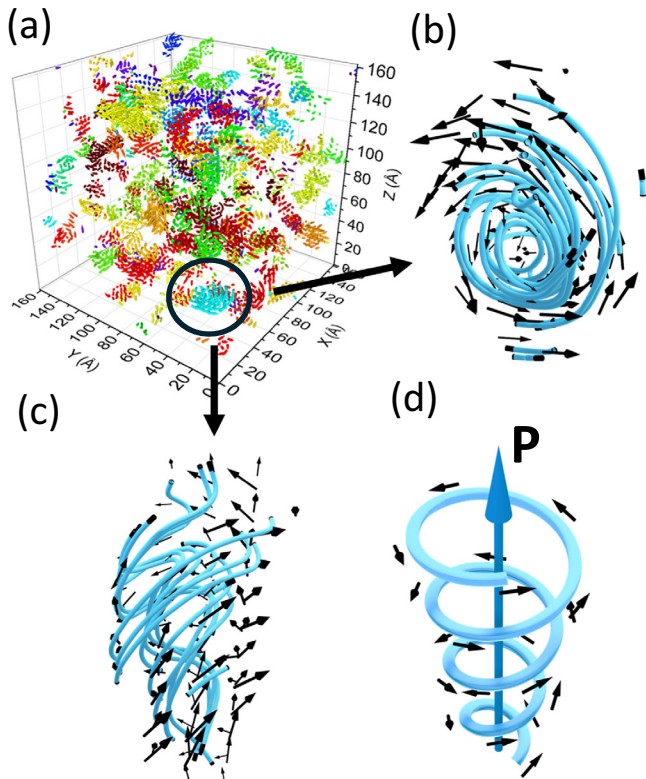

**Fig. 4 | Swirling textures of polarization and merons.** Polarization vortices identified in the PMN configuration at 300 K refined using the experimental data in Figure. S3, S4, S6, and S7. **a** 3D rendering of Pb displacement (filtered) vector field displaying the largest vortex-like clusters. **b**, **c** different views of the Pb displacement pattern in one of the well-defined meron-like configurations in (**a**). Lines are streamlines introduced as a guide to the eye. The tube axis here is approximately parallel to the [011] direction. **d** Schematic rendering of a meron tube with the blue arrow indicating the direction of the net polarization, **P**.

[TiPb$_8$] clusters, resulting in a broad and distorted distribution of Ti-Pb distances.

### Inter-PNR correlations and swirling textures of polarization

A striking feature of the projected filtered Pb displacements is that they form well-defined vortices (Fig. 2b). Similar patterns are observed in the local polarization vectors derived directly from the refined atomic coordinates (Fig. 2c). We calculated various metrics of vorticity (i.e., curl, $Q$-criterion, etc., see Methods) for the 3D displacement field to identify the vortex-like features. Figure 4 illustrates a cluster of Pb displacements that spiral around the ⟨011⟩ direction, resembling a meron tube with a net displacement along its axis[29]. We also observed similar winding textures for the B-cation displacements (Fig. S27), consistent with the strong correlations between the two sublattices.

Meron-like formations in PMN (Fig. 4) can be viewed as consisting of correlated PNRs, predominantly pairs with the ⟨111⟩ and ⟨001⟩ displacements (angle ≈ 54°) or 71° ⟨111⟩ variants (Fig. S29). A previous study[10] identified a hierarchical structure, highlighting a fragment where several 71° PNRs create a curling polarization, but did not elaborate on the nature of the observed hierarchy. Our results here demonstrate how such "elementary" PNRs, referred to as α-PNRs in ref. 10, organize into topological textures. Streamlines of the displacement vector field reveal a complex correlation topology characterized by the continuously curling polarization and multiple vortices identifiable by larger curl-metric values (Figs. 2b and 4). Using this metric and HDBSCAN[30], we isolated atoms that form vortex cores, which were

predominantly found near the boundaries separating the nanoscale regions with the net positive and negative charges (Fig. 1c) arising from the B-cation distribution. This relationship is visible in 2D slices of the refined configuration (Fig. 5a, b) and is supported by a statistical analysis of the distances between the core centers and charge boundaries in three dimensions (Fig. 5c, d). Thus, the vorticity of the displacement field peaks at the maxima of the electric charge gradient, which is defined by the local chemistry.

The DS calculated for the Pb atoms with the largest curl metric appeared as isotropic blobs around the Bragg peaks, modifying the intensity dependence on **q**. We could not distinguish between the scattering from the spiraling pattern and other types of correlations. Nonetheless, this result highlights the importance of accurately reproducing the diffuse intensity behavior to capture all the correlations in the displacement field.

The experimental data used in this study and our refined models represent structural snapshots (see methods). Energy-resolved neutron scattering measurements in PMN[31] (≈0.5 μeV energy resolution) show that above 420 K, the elastic DS mostly disappears. Hence, at 490 K, the DS appears to be mainly dynamic in origin, while at 300 K, the static contributions to the diffuse intensity become significant, albeit still non-dominant[31,32]. The characteristic decay times associated with these dynamics were determined to be of the order of $10^{-1}$ ns to $10^{-2}$ ns[31]. We thus conclude that the swirling topology of polarization identified here is essentially dynamic, albeit influenced by chemistry, as the locations of the vortex cores are tied to the B-cation distribution. The observed overlapping correlation volumes for the components of polar displacements are consistent with the phenomenological model of a slowly changing displacement pattern resembling the result of thermal vibrations, which was proposed in refs. 26,33. It would be interesting to compare the diffuse scattering shown for structural projections in Fig. 2 with that calculated for analogous projections obtained using atomic-resolution STEM images[11,18,27]. While projections of our configurations represent structural snapshots, those from STEM highlight purely static displacements.

The topological ordering of polarization in ferroelectrics is of significant fundamental and technological interest[34]. Vortices and bubble domains have been observed in thin-film heterostructures[34] and nanoparticles[35], where they minimize large depolarization fields. Levanyuk and Blinc conjectured that bulk relaxors could also undergo a local phase transition to a vortex state[36]. Recent atomic-resolution STEM imaging has provided evidence of bubble domains with skyrmion topologies in bulk NBT-based ceramics, and concurrent phase-field modeling suggested that these topological textures arise from a balance of bulk, elastic, and electrostatic energies among co-existing nanoscale regions of distinct symmetries[37]. Simulations for bulk PZT also indicated that the topological ordering of PNRs can reduce depolarizing fields[38]. Our finding of swirling polarization textures containing a high number density of vortices in bulk PMN-PT not only reinforces the theoretical predictions but also demonstrates the very existence of this phenomenon. The localization of vortices at the charge boundaries created by the nanoscale chemical inhomogeneities observed here suggests that these boundaries act similarly to interfaces or surfaces in synthetic ferroelectric nanostructures, creating depolarizing fields that drive the development of topological textures and defects. In addition, these compositional variations introduce strain gradients that, via the flexoelectric coupling[39], have been shown to influence topological structures in ferroelectrics[40].

### Discussion

We used an integrative structural refinement framework and a novel software tool to identify the elusive interatomic correlations in PbMg$_{1/3}$Nb$_{2/3}$O$_3$-PbTiO$_3$ (PMN-PT) relaxor ferroelectrics. The previously proposed hierarchical assembly of polar nanoregions (PNR),

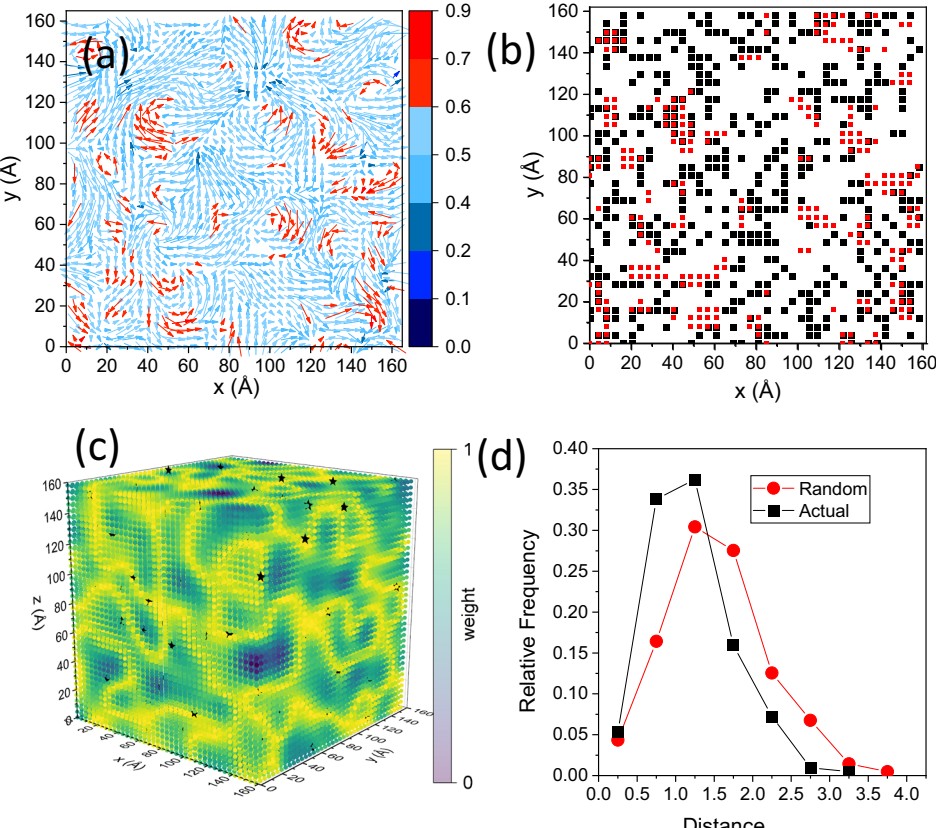

**Fig. 5 | Pinning of polarization vortices by the local chemistry. a** Single atomic layer of Pb in PMN, with vectors indicating projections of the filtered Pb displacements onto the layer's plane. Color – Q-criterion of vorticity. The color scale is selected to emphasize the displacements with Q > 0.6. **b** An overlay of the Pb atoms having Q > 0.6 (red symbols) and locations of the electric charge boundaries (black symbols) in this layer arising from the distribution of the neighboring Mg and Nb. This image reveals a clear preference for the Pb atoms acquiring large Q values to be located in the vicinity of these boundaries. **c** 3D rendering of the charge boundaries (highlighted in yellow) in the refined configuration of PMN, with the asterisks marking the centers of Pb clusters having Q > 0.6 (identified using HDBSCAN). **d** Statistical distributions of distances from the centers of clusters with large vorticity to the charge boundaries. Black−cluster centers as found in the refined configuration. Red−randomized locations of the cluster centers. The Matt-Whitney test confirms that the actual distances are significantly shorter than those for randomly distributed cluster centers, in line with the visual inference from Fig. 5b.

dominated by the local Pb displacements, has been revealed as overlapping volumes with anisotropic shapes having one principal axis longer than the others. Each volume encompasses atoms with projections of their displacement vectors onto this volume's long axis having the same sign – a correlation that determines the characteristic anisotropy of the observed DS near the Bragg peaks, a hallmark of the PMN-PT system. The overlap of these volumes creates a 3D patchwork of PNRs, with the displacements in the overlapping spaces being a vector sum of the components. The polarization follows these displacement patterns. The nanoscale chemical ordering of the octahedral cations is accompanied by the segregation of Mg into regions with stronger ordering, which enhances the nanoscale compositional heterogeneities. Our results indicate that the compositional effect on polar displacements dominates over that of the ordering. Volumes exhibiting large net polarization that contribute the most to the observed DS occur preferentially in Mg-deficient regions, signifying the disruptive role of Mg in establishing polar correlations. The Pb displacements and polarization vector fields are also tied to the chemistry via a divergence that changes from positive to negative between the Nb-rich and Mg-rich regions.

While we invoked PNRs to describe the observed swirling textures of polarization, it was done as a matter of convenience rather than a necessity, similar to a way of rationalizing some topological textures using flux closure domains[41,42]. However, applying the PNR concept obscures the continuous nature of the changes in the direction and magnitude of polarization, where using vector-field descriptors (e.g., divergences, curl, α-metric) could have been more appropriate. That is, the answer to the question posed by Hlinka[12] regarding the need for the ether of polar nanoregions appears to be no; we do not. Takenaka et al.[9]. drew an analogy between the thermal evolution of relaxors and water, which turns into a slush upon cooling, before freezing. The PNRs were suggested to act like bits of ice in this slush, able to rotate independently from one another. However, we observe a distinctly different scenario where the polar structure is a continuous interwoven maze.

The picture uncovered here shows the polarization vectors aligned over the nanoscale, oscillating around several preferred crystallographic directions. The latter are maintained over multiple unit cells before "branching" to symmetry-equivalent or other favorable directions. This branching yields smooth changes in the polarization direction, creating swirling textures. On the basis of prior simulations[37,38] and our own observations of extended strain and charge heterogeneities, we hypothesize that this swirling topology simultaneously minimizes strain and electrostatic energies. The oscillations and branching of the nanoscale polarization directions are presumed to arise from the chemical heterogeneities discussed above, also minimizing the polarization-gradient energy. Indeed, the vortices' cores, where the vector curl magnitude is the largest, are pinned by the local chemistry, being located preferentially near the boundaries separating the negatively (Mg$^{2+}$-rich) and positively (Nb$^{5+}$-rich) charged

regions. When subjected to an AC electric field, these intertwined topological textures should exhibit broader relaxation-time distributions than isolated PNRs (e.g., cases with high concentrations of the "blocking" species). This could contribute to a significantly broader dielectric response observed in PMN compared to the relaxor systems, e.g., Ba(Zr, Ti)$O_3$-based, which incorporate mixtures of isovalent species that do not generate depolarizing fields. The sizes of the vortices and their distribution likely depend on the balance of multiple energy terms (elastic, electrostatic, polarization-gradient, flexoelectric). Understanding the relative importance of these terms will require targeted simulations.

A transition from relaxor to classic ferroelectric behavior in PMN-PT aligns with a basic crystal-chemical framework[22], in which the type and concentration of blocking ions influence the correlation length of polar distortions. As the chemical ordering becomes significantly weaker and more limited in scale, the accompanying segregation of Mg is also diminished. The combination of aliovalent $Mg^{2+}$, $Ti^{4+}$, and $Nb^{5+}$ still results in extended charge fluctuations; however, such fluctuations are considerably smaller in magnitude compared to those in PMN (Fig. S30). This reduction lowers the barriers to establishing long-range ferroelectric ordering upon cooling. Our analysis indicates that a largely disordered mixture of Ti, Nb, and Mg still affects the directions of lead (Pb) displacements. For instance, Ti behaves similarly to Mg by promoting a negative divergence in the Pb vector field, in contrast to the positive divergence associated with Nb. This influence of the local chemistry may explain why the correlated polar disorder observed in the cubic ergodic state persists in the lower-symmetry monoclinic and tetragonal structures of PMN-PT. We anticipate that continuous polarization textures observed here may also emerge in other systems where nanoscale chemical inhomogeneities give rise to strain gradients and internal charge fields.

## Methods

### Sample Synthesis

Powder specimens of (1-$x$)PMN-$x$PT with $x$ = 0.3, 0.35 were synthesized using the columbite route described in ref. 43. Powders of $MgCO_3$ and $Nb_2O_5$ (with 2 mol % excess MgO) were reacted at 900 °C for 8 h in air to form $MgNb_2O_6$, followed by another heating at 1200C for 12 h. This product was batched with PbO (5 wt. % excess) and $TiO_2$ (<10 ppm $P_2O_5$) and heated in a covered crucible at 850 °C for 4 h and then at 1000 °C for 2 h. The cover was sealed using alumina-based paste to minimize PbO loss. Prior to each heating, the powders were mixed and ground in a planetary ball mill in isopropanol using yttria-stabilized zirconia grinding media. Single crystals of these compositions were produced using a flux-growth technique, which exploits spontaneous nucleation by gradually cooling a supersaturated solution from high temperatures.

### Data collection and reduction

The phase purity of the powder specimens was confirmed using a laboratory X-ray powder diffractometer equipped with Cu K$\alpha_1$ radiation. Variable-temperature neutron total scattering measurements on powder specimens were performed using POLARIS diffractometer at ISIS. The sample powders were loaded in 6 mm vanadium cans. Temperature control was achieved using a furnace. For PMN ($x$ = 0), we used data collected on this instrument at 300 K and 490 K previously[10]. For the PMN-PT compositions, total scattering data were collected at 300 K, 383 K, and 533 K, with these temperatures selected to probe the different phase fields. Diffraction data with Bragg peaks suitable for Rietveld refinements were collected at intermediate temperatures. X-ray total scattering data were collected at the same temperatures using the I15-1 beamline at the Diamond Light Source with the incident X-ray energy of 76.6 keV and an amorphous-silicon 2D detector about 20 cm downstream from the sample. The sample powders were loaded in fused quartz

capillaries, 1 mm in diameter. Temperature was controlled using cryostream and, above 500 K, with a hot-air blower. We collected data from the Si NIST SRM to calibrate the instrumental resolution function for the neutron and X-ray experiments. The neutron total scattering data were reduced using the GUDRUN software to extract the neutron scattering function, S($Q$), and its Fourier transform. The X-ray total scattering data were corrected for parallax effects and then reduced in the PDFGetX3 software[44] to extract the X-ray scattering function.

Variable-temperature extended X-ray absorption fine structure (EXAFS) measurements were performed for the Pb $L_3$, Nb $K$, and Ti $K$ edges using the NIST 6-BM beamline at NSLS-II (Brookhaven National Laboratory). The data were collected in transmission for Pb and Nb and fluorescence for Ti. The latter was recorded using a 4-element Si-drift detector. Temperature control was achieved using a contact heater with sample powders deposited on a 10-micron-thick aluminum foil. The X-ray absorption spectra were reduced in the Athena software[45] to extract the EXAFS signal. Preliminary fitting of the EXAFS data was performed in Artemis[45], with the scattering amplitudes and phases for a photoelectron calculated using self-consistent muffin-tin potentials in FEFF8[46].

Single-crystal X-ray DS was measured using the beamline ID28 of the European Synchrotron Radiation Facility. Thin, rod-like crystals were mounted on a rotation stage with a hot-air blower used for temperature control. Before the measurements, the samples were etched using hot, concentrated hydrochloric acid. DS patterns were recorded with a wavelength of 0.954 Å over the angular range of 360° in 0.1° increments on a PILATUS 1 M (Certain equipment, instruments, software, or materials, commercial or noncommercial, are identified in this paper in order to specify the experimental procedure adequately. Such identification is not intended to imply recommendation or endorsement of any product or service by NIST, nor is it intended to imply that the materials or equipment identified are necessarily the best available for the purpose.) single-photon counting pixel detector. The CrysAlisPro software package was used to refine the orientation matrix, and the final reciprocal-space reconstructions of 3D diffuse-intensity distributions were accomplished using ID28-based computer software. The measured diffuse scattering was symmetrized using operations of m-3m point symmetry group to maximize the reciprocal-space coverage and mitigate possible X-ray absorption effects.

Single-crystal neutron DS data were collected on CORELLI spectrometer at the Spallation Neutron Source (SNS). These crystals were physically different from those used in the X-ray measurements, but were grown in the same laboratory under similar conditions. The crystals were measured at the same temperatures as used for other data. Temperature control was achieved by placing the samples in a cryo-furnace. Data processing to determine the orientation matric and obtain 3D reconstructions of the scattered intensity was performed using custom scripts developed at SNS. CORELLI is equipped with a statistical correlation chopper, which permits separating elastic scattering (energy resolution ≈1 meV)[47]. The 3D reconstructions were obtained with and without this chopper to compare the elastic and energy-integrated diffuse intensities. Fitting of diffuse intensity near Bragg peaks in energy-integrated 3D datasets is currently precluded by the resolution profile effects, causing extended asymmetric streaks of intensity along **Q** and towards the origin for all Bragg peaks. An additional complication is diffraction from polycrystalline aluminum foil used to shield the sample for temperature stability, which is manifested as spherical shells of intensity with the radii corresponding to the $Q$-values for the aluminum Bragg reflections.

### Structural refinements

Rietveld refinements using the neutron powder diffraction data were performed in the TOPAS software[48] to obtain average structure characteristics. This report focuses on the high-temperature cubic phases

where the only refined structural variables were lattice parameters (Fig. S1) and atomic displacement parameters.

A principal challenge in fitting the intensity of single-crystal DS is placing it onto the absolute scale. Bragg peaks recorded with DS are usually saturated and unsuitable for intensity calibration. The scale assigned to the diffuse intensity influences the strength of determined atomic correlations and, in the case of displacements, their magnitudes. We addressed this issue by developing a relatively rigorous and robust procedure for scaling the diffuse intensity, which is expected to yield significantly more accurate results than previously possible. In this approach, we perform spherical integration of an experimental 3D DS dataset to obtain a one-dimensional trace of intensity versus the modulus of the scattering vector $\mathbf{Q}$. The resulting $I(Q)$ signal is compared to that obtained via a similar spherical integration of a 3D diffuse intensity distribution calculated for a sufficiently large atomic configuration. In this configuration, the atoms are displaced randomly according to atomic displacement parameters determined using Rietveld or single-crystal refinements from Bragg-peak intensities. The calculated intensities are normalized by the number of atoms in the configuration. We expect the experimental and calculated $I(Q)$ traces to match at large $Q$ values (Fig. S2a). Therefore, their ratio provides an estimate of the scale factor that should be applied to the experimental data. As an additional check, after rescaling and normalizing the intensity by the average scattering factor squared, we anticipate the experimental (diffuse component) and calculated traces to match the baseline in the total scattering function obtained from powder samples (Fig. S2b).

RMC refinements were performed in the RMCProfile software[10,49–52] using simultaneous fitting of the neutron and X-ray total scattering data in their reciprocal and real-space representations, neutron Bragg profile, EXAFS (Pb and Nb for PMN and Pb, Nb, and Ti for PMN-PT), and 3D reconstructions of the diffuse intensity scaled as described above (Fig. S3 through Fig. S13, also Fig. S26). The structures were represented using atomic configurations of $40 \times 40 \times 40$ perovskite cubic unit cells, which span distances up to $\approx 16$ nm. For the DS dataset, we used a grid with $40 \times 40 \times 40$ voxels per reciprocal space volume bound by $H = 1$, $K = 1$, and $L = 1$. Matching the number of voxels with the number of unit cells in the configuration was required for a Lanczos filter implemented to smooth the otherwise too-noisy calculated diffuse intensity. Unlike the previously used, more economical but aggressive, smoothing approach, the new procedure provides adequate smoothing[53] without concurrent broadening of the diffuse features, which may impact correlation lengths in the refined configuration. The DS datasets covered half of the reciprocal space with $L \geq 0$. Because of computing speed limitations, we could only include the diffuse intensity out to $H = K = \pm 2.2$, $L = 2.2$ in the fits. In addition to the experimental data, we applied restraints that favored the least distorted oxygen octahedra with the target metal-oxygen distances set for each B-cation species separately according to their ionic radii and bond valence criteria. In the refined configurations, the resulting distances differed from these idealized values, being determined by the EXAFS data. Still, these restraints helped to regularize the configuration. During refinements, weights assigned to individual datasets and restraints are selected automatically following the procedure outlined in ref. 53, which performs statistical analysis of changes in individual residual components as a function of atomic moves.

Refinement proceeded according to the following strategy. Initial configurations were generated by starting with random distributions of the octahedral metal cations (Nb and Mg for PMN and Nb, Mg, and Ti for PMN-PT), and all the atoms, including Pb and O, were displaced randomly according to their ADP values (anisotropic for oxygen) obtained from our Rietveld refinements. The distributions of the B-cations in these configurations were refined in the RMCProfile software by swapping the distinct cation species among their sites while

fitting the DS intensity at and around the reciprocal space $R$ points with $H$, $K$, and $L$, all half integers, which are known to reflect the short-range order of these cations. No displacement moves were allowed at this stage. Once these swap-move refinements converged, fits of the same diffuse features continued, allowing displacements of the oxygen atoms (all other atoms were left unchanged) under the polyhedral restraints to permit octahedral relaxations in response to the changes in the cation distribution. For the PMN-PT compositions, the DS contained only the first-order ½ ½ ½ superlattice reflections related to the cation ordering. These reflections were much weaker and broader compared to PMN, and they were insufficient for refining the distributions of the three B-site species. Therefore, for the PMN-PT compositions, we combined both X-ray and neutron DS datasets, fitting the volumes of the reciprocal space that contained the superlattice reflections and their surrounding background.

For PMN, we performed independent refinements of chemical short-range order for the data collected at 300 K and 490 K and obtained similar order parameters for both cases. We then used the configuration for 490 K as a starting point for refining the 300 K structure. Refining the chemical order first, instead of allowing all the atoms to move while swapping the B-cations, is to avoid artifacts with the shifts of the heavy Pb and Nb species, which dominate X-ray scattering, creating an ordered pattern of displacements that reproduce the diffuse intensity. Previously, we allowed for the Mg and Nb swaps and atomic shifts during later stages of refinements, but it didn't change the outcome. As shown in ref. 10, the cation ordering obtained under this approach is in good agreement with atomic-resolution STEM images[54], attesting to the adequacy of the assumption that attributes the intensity of the ½ ½ ½ diffuse reflections to the distribution of Nb and Mg and the associated oxygen relaxations. For each composition, several starting models with different distributions of the B-cations were generated following the above procedure and then used as starting models for refining atomic displacements.

A potentially limited energy integration window in the neutron total scattering measurements has been argued to be wide enough to encompass at least the low-frequency modes involving the Pb off-centering[10].

### Analyses of refined atomic configurations

The octahedral cation distributions were characterized using the Warren-Cowley[55] short-range order parameters and the local metric introduced in ref. 10. This local metric, $\eta$, is defined as $\eta = (n_1 - n_2)^2 + (n_4 - n_3)^2$. Here, $n_i$ is the fraction of a given type of species (e.g., Nb) found in the $i^{th}$ coordination shell around a particular B-cation (Nb or Mg). The $n_i$ fractions are normalized by the values expected from the average composition. The $\eta$ metric was calculated for each B-cation in the configuration. With this definition, clusters of the B-cations with large values of $\eta$ represent chemically ordered regions.

Distortions of the oxygen octahedra, including their rotations as rigid units, were determined using a set of 18 orthonormal deformation modes described in ref. 10. The degree of ordering for octahedral rotations was quantified using the same Warren-Cowley-type order parameter, $\kappa$, as introduced in ref. 56: $\kappa(\iota) = (p(\iota) - 0.5)/0.5$, where $\iota$ labels the coordination shell, in the plane normal to the rotation axis, around a "reference" octahedron, and $p(\iota)$ is the probability that an octahedron in shell $\iota^{th}$ is rotated opposite to the reference. This order parameter reveals the existence and spatial extent of a cogwheel pattern of rotations[56] in individual octahedral layers. Note that $\kappa(\iota)$ only captures intralayer ordering; describing rotations of octahedra along the rotation axis requires an additional order parameter.

Correlated components of atomic displacements contributing to the DS in the refined configuration were extracted via the Fourier transform of the DS complex amplitude calculated from the refined

atomic coordinates (Fig. S31). The regions of reciprocal space containing the diffuse intensity of interest were selected using spherical masks, 0.2 Å$^{-1}$ in diameter. For the DS near Bragg peaks, all peaks out to $H = K = L = 20$ ($Q \approx 31$ Å$^{-1}$) were included—this wide coverage is computationally expensive but required to provide sufficient real-space resolution. In real space, the transform was calculated on a grid with a mesh size of 0.05 Å. The Fourier transform is real and features positive and negative peaks around atomic positions. Atoms that acquire the largest intensity of these peaks contribute the most to the DS of interest. Generally, there may be several positive peaks of varying intensity near an atomic position. A vector that connects the centers of mass for the intensity-weighted positive and negative peaks, respectively, represents the component of a displacement for a given atom that participates in the correlations, generating the diffuse intensity included in the transform. The Fourier filtering of the diffuse amplitude was accomplished using a newly developed software optimized to enable these calculations over a large volume of reciprocal space and for a large number of atoms in the configuration, as required to reveal a complete set of correlated displacements. Figure S31 illustrates the ability of the procedure to recover correlated atomic displacements for a simulated model containing three orientational domain variants.

## Calculations of the local polarization

The deviation of the polarization for the $i^{th}$ unit cell from the configurational average was calculated (for the center of the unit cell placed at a B-cation) as

$$\Delta P_i = Q_{ij}^B \Delta r_j^B + \frac{1}{2}\sum_{n=1}^{6} Q_{ij}^{O_n} \Delta r_j^{O_n} + \frac{1}{8}\sum_{n=1}^{8} Q_{ij}^{A_n} \Delta r_j^{A_n}$$

where $Q_{ij}^B$, $Q_{ij}^{O_n}$, and $Q_{ij}^{A_n}$ are components of the Born effective charge (BEC) tensors for the central B cation and its O and A-cation neighbors, respectively, while $\Delta r_j^B$, $\Delta r_j^{O_n}$, and $\Delta r_j^{A_n}$ are the displacements of these ions from their respective average positions in the configuration. The formula is modified accordingly if the center is placed on an A-cation. For a cubic average structure, $\Delta P_i$ represents a unit-cell polarization, $P_i$. However, its value depends on the choice of the cell center (i.e., B or A cation). Therefore, we calculated the average $P_i$ over a sampling volume of $N \times N \times N$ unit cells, sliding this volume over the configuration as in the boxcar method. The differences associated with selecting particular cation species as the center cancel already for $N = 2$, facilitating the analysis. Following the trends in the average modulus of the local polarization as a function of $N$ highlights the correlation length for the polarization, characterized by the number of cells required to attain the macroscopic (configuration) average. For PMN, BEC tensors have been determined using first-principles calculations[57]; however, the BEC values vary with the type of ordering for Mg and Nb. For PMN-PT, no BEC charges are available in the literature. Here, we are interested not in the absolute magnitude of the polarization but in the patterns of the local polarization vectors and the evolution of the polarization magnitude with the sampling volume. We compared the behavior of the local polarization in PMN calculated assuming the BEC values determined from first principles calculations for the [001]-type ordering (Pb – 4, Nb – 7.4, Mg – 2.6, O – –4.8 and –2.5 parallel and perpendicular to the B-O bond, respectively) and that for the polarization calculated assuming formal ionic charges. We found no significant differences in the patterns of the polarization displacement vectors for the two cases. Therefore, we adopted the formal charges to facilitate the comparison between the behavior of the local polarization in PMN and PMN-PT solid solutions.

Figure 2d displays trends in the modulus of the average local polarization, $\bar{P}_N$, as a function of the sampling volume, defined as a box of $N \times N \times N$ unit cells. In our configurations, the average polarization, $\bar{P}_{40} = 0$. For a given $N$, the $\bar{P}_N$-value is obtained by averaging the moduli

of the polarization within the sampling box, calculated while sliding this box over the configuration. As a reference, we included a trend for a configuration of PMN at 300 K with the refined atomic displacement vectors scrambled over the positions of each species. For this random case, the average local polarization magnitude approaches zero at $N \approx 15$. For the actual configuration of PMN at 300 K, the macroscopic average is attained for $N$ between 35 and 40. The decay rate for the $\bar{P}_N$ vs. $N$ is significantly slower than in the example with random displacements, indicative of extended polar correlations. For PMN at 490 K and the PMN-PT configurations, the $\bar{P}_N$ falls off considerably faster but still slower than in the random case.

## Analyzing octahedral deformations and rotations

Octahedral deformations, including rotations, were characterized using the previously developed deformation mode analysis[10]. In this method, the distortions of each octahedron are described using 18 orthonormal deformation modes, which are selected to represent typical lattice distortions and vibrational modes (see Figure. S15 in ref. [10]). In PMN, [MgO$_6$] octahedra expand relative to the average, whereas [NbO$_6$] octahedra contract, as expected from the differences in the ionic radii of Mg and Nb. A strong positive correlation (Pearson correlation coefficient 0.2) exists between the local chemical ordering parameter and the magnitude of this "breathing mode," which is consistent with the regions of stronger chemical ordering permitting larger oxygen shifts along the Mg-O-Nb bonds as required to satisfy the bonding requirements of both Mg and Nb without an overall strain. Likewise, in the PMN-PT configurations, the [MgO$_6$] octahedra expand, whereas both [NbO$_6$] and [TiO$_6$] octahedra contract, with the much more significant shrinkage of the latter consistent with the Ti ionic radius being smaller than that of Nb.

In PMN, analysis of octahedral tilting modes around the cubic axes reveals a cogwheel pattern of rotations ordered over a short range within octahedral layers normal to the corresponding rotation axis (Fig. 1D). The average rotation angle about each cubic axis is about 4°, and the correlation is limited to the first two coordination shells. At 490 K, the correlations are significant only for the nearest-neighbor octahedra. Analysis of correlations along the octahedral chains parallel to the rotation axes highlights a weak preference for the in-phase tilts. That is, the PMN structure at 300 K appears to contain small clusters, 1–2 nm in size, with $a^0 a^0 c^+$-type tilting. At 490 K, the cluster size is reduced to just the neighboring octahedra. The magnitude of octahedral rotations correlated positively with the local chemical order parameter for the cations; however, the chemical and in-plane tilting order parameters appeared uncorrelated. No ordering of rotations was detected for the PMN-PT compositions.

## Vector-field metrics

### Q-criterion

Hunt et al.[58]. have proposed this metric as effective in distinguishing regions of coherent rotation motion from those dominated by strain. While originally developed for fluid flows, it is applicable to displacement vector fields for identifying vortex-like or rotational structures.

We denote a displacement vector field as $u(x,y,z) = (u_x(x,y,z), u_y(x,y,z), u_z(x,y,z))$, where $u_x, u_y, u_z$ are the components of the vector field along the x, y, and z axes, respectively.

The spatial gradient of $u$, $\nabla u$, is a second-order tensor:

$$\nabla u = \begin{pmatrix} \frac{\partial u_x}{\partial x} & \frac{\partial u_x}{\partial y} & \frac{\partial u_x}{\partial z} \\ \frac{\partial u_y}{\partial x} & \frac{\partial u_y}{\partial y} & \frac{\partial u_y}{\partial z} \\ \frac{\partial u_z}{\partial x} & \frac{\partial u_z}{\partial y} & \frac{\partial u_z}{\partial z} \end{pmatrix}$$

This tensor captures all first-order spatial derivatives of the vector field, combining information about both deformation and rotation.

The gradient tensor can be decomposed into a symmetric strain-rate tensor $S$ and a skew-symmetric rotation tensor $\Omega$, defined as

$$S = \frac{1}{2}\left(\nabla u + (\nabla u)^T\right)$$

$$\Omega = \frac{1}{2}\left(\nabla u - (\nabla u)^T\right).$$

Then, the Q-criterion is defined as a local measure of excess rotation relative to strain:

$$Q = \frac{1}{2}\left(||\Omega||^2 - ||S||^2\right)$$

If $Q > 0$, rotation dominates over strain, indicating a vortex or vortex-like region. Alternatively, if $Q < 0$, strain dominates.

Curl:

The curl is another metric describing the tendency of a vector field to exhibit the local rotation. It is a vector having a direction aligned with the axis of rotation and a magnitude reflecting the rotational strength. The curl is defined as:

$$\nabla \times u = \left(\frac{\partial u_z}{\partial y} - \frac{\partial u_y}{\partial z}, \frac{\partial u_x}{\partial z} - \frac{\partial u_z}{\partial x}, \frac{\partial u_y}{\partial x} - \frac{\partial u_x}{\partial y}\right)$$

**Divergence.** The divergence measures the flow of the vector field from a given point. It is a scalar field defined as:

$$\nabla \cdot u = \frac{\partial u_x}{\partial x} + \frac{\partial u_y}{\partial y} + \frac{\partial u_z}{\partial z}$$

## Data availability

The experimental data and all the files necessary for performing the structural refinements in RMCProfile for the three compositions reported here have been deposited in the MIDAS at NIST, https://doi.org/10.18434/mds2-3871.

## Code availability

The RMCProfile software (executables) can be downloaded from https://rmcprofile.ornl.gov. The MOSAIC software for the Fourier filtering of the diffuse scattering amplitudes is being prepared for public release.

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

## Acknowledgements

Experiments at the ISIS Pulsed Neutron and Muon Source (RB 1920053) were supported by beamtime allocation from the UK Science and Technology Facility Council. We thank Diamond Light Source for access to beamline I15-1 (proposal CY27040) and the European Synchrotron Radiation Facility for the provision of beam time on ID28. Portions of this research were carried out at the National Synchrotron Light Source II (NIST beamline 6-BM) and at the Spallation Neutron Source, operated for the DOE Office of Science by Brookhaven National Laboratory and the Oak Ridge National Laboratory under Contracts No. DE-AC02-98CG10886 and DE-SC0012704, respectively. The beamtime at SNS was allocated to CORELLI on proposal number IPTS-32622.1. S.G. and I.L. acknowledge the US-Israel Binational Science Foundation for their financial support (Award No. 2018161). S.G. was also supported by the Israel Science Foundation (Awards 1561/18, 1365/23). B.-X.W. and Z.-G.Y. acknowledge the support from the Natural Sciences & Engineering Research Council of Canada (DG, RGPIN-2023-04416) and the U.S. Office of Naval Research (N00014-21-1-2085). This manuscript has been authored by UT-Battelle, LLC, under contract DE-AC05-00OR22725 with the US Department of Energy (DOE). The US government retains and the publisher, by accepting the article for publication, acknowledges that the US government retains a nonexclusive, paid-up, irrevocable, worldwide license to publish or reproduce the published form of this manuscript, or allow others to do so, for US government purposes. DOE will provide public access to these results of federally sponsored research in accordance with the DOE Public Access Plan (https://www.energy.gov/doe-public-access-plan).

## Author contributions

Conceptualization: I.L., S.G., and M. G.T. Synthesis: W. J. L., B.-X.W., and Z-G.Y. Data acquisition and processing: I. L., S.G., H.Y.P., A. B., M.G.T., B. R., P. A. C., F. Y., and A.M. Methodology: M. E., V.K., and I.L. Software development: M.E. and V.K. Data analysis: M. E., V. K., and I.L. Super-vision: M.G.T. and I.L. Writing—original draft: I.L. and M.E. Writing—review and editing: All coauthors.

## Competing interests
The authors declare no competing interests.

## End Notes
In crystalline solid solutions, even a random distribution of species sharing the same crystallographic sites leads to intrinsic nanoscale clustering of unit-cell characteristics, such as composition, charge, strain, etc. In PMN-PT, a structure with a random distribution of Mg and Nb or Mg, Nb, and Ti would exhibit extended regions with net positive and negative electric charges. The chemical ordering tends to increase the size of these regions.
