## [Transparent Peer Review file · Nature Communications]

Emergent topological polarization textures in relaxor ferroelectrics

Corresponding Author: Dr Igor Levin

Version 0:

Reviewer comments:

Reviewer #1

(Remarks to the Author)

This paper reconstructs 3D mesoscale polarization maps in the classic relaxor system $\text{PbMg}_{1/3}\text{Nb}_{2/3}\text{O}_3\text{-PbTiO}_3$, indicating that chemical disorder stabilizes inner polar topologies, offering a route for engineering new dielectric and ferroelectric functionalities. I suggest that could be accepted after addressing the following issues:

1. The author focused on the PMN-30PT and PMN-35PT, can this be observed in other ratio doping or material doping?
2. It is suggested to add more experimental results to give more convincing conclusion.

Reviewer #2

(Remarks to the Author)

The manuscript by M. Eremenko et al. presents a new view on the polar structure of relaxor ferroelectrics. Based on simultaneous fitting of complementary x-ray and neutron total scattering, x-ray absorption spectroscopy, and diffuse scattering, they reconstruct polarization maps for $\text{PbMg}_{1/3}\text{Nb}_{2/3}\text{O}_3$ (PMN) and $\text{PbMg}_{1/3}\text{Nb}_{2/3}\text{O}_3\text{-PbTiO}_3$ (PMN-PT) with 30 and 35 at. % PT. From the analysis of these data, they proposed a picture of polarization vectors organized in swirling textures to simultaneously minimize stresses and depolarizing fields. Oscillations and branching of the local polarization directions are linked to the nanoscale chemical heterogeneities with the vortices' cores located preferentially near the boundaries separating the negatively and positively charged regions. This picture differs from the polar nanoregions model and the "slush-ice" model of relaxors.

The paper makes an important step in understanding the relaxors puzzle and will be of interest and importance to readers working in the field of disordered materials. The manuscript is very well written. The discussion and conclusions sound good.

I recommend it for publication in Nature Communications. I have a small remark.

1. At the end, the authors write about a difference between their picture proposed for PMN-based and other relaxor systems, such as isovalent $\text{BaTiO}_3\text{-BaZrO}_3$ solid solutions. Therefore, it is proposed to change the title to "Emergent topological polarization textures in $\text{PbMg}_{1/3}\text{Nb}_{2/3}\text{O}_3\text{-PbTiO}_3$ relaxor ferroelectrics."

Reviewer #3

(Remarks to the Author)

The authors employed an integrated structural refinement framework, combining X-ray/neutron total scattering, X-ray absorption spectroscopy, and diffuse scattering data, to reconstruct three-dimensional polarization maps in the PMN-PT system. This multi-technique collaborative approach breaks the limitations of single scattering data, providing a new paradigm for the analysis of complex structures. This work reveals the presence of self-organized swirling polarization textures in relaxor ferroelectrics PMN-PT, challenging the conventional PNRs model. This finding expands the understanding of polar inhomogeneities and promotes the development of relaxor ferroelectric theory. While the data analysis is relatively thorough, several aspects require further clarification and modification.

1. The current refinement procedure employs a sequential approach, first optimizing B-site cation ordering through atom

swaps before allowing O atom displacement. The author should justify why they do not employ simultaneous refinement (e.g., using SWAP_MULTI) allowing displacement and swap. An analysis of results obtained from both approaches would be valuable to demonstrate the rationality of this refinement strategy.

2. Fig.1 presents three distinct order parameters, but the calculated procedures require further clarification, especially the local metric order parameter (Fig.1(a)), and rotational order parameter (Fig.1(d)). Moreover, Figure 1(a) only shows for one layer Pb, and is this same for different layers?

3. The definition of average angle needs further clarification. Fig.2(b) used the Fourier filtering of the calculated DS amplitude, and any other data included (e.g., total scattering, EXAFS)? If included, what are the weighting factors for these data sets?

4. The "Correlations among polar displacements: correlation length vs. PNR size" part (particularly from line 204) requires reorganization to clearly distinguish the contents of Fig.2 and Fig.3.

5. The author pointed that the λ -PNR is smaller than the correlation length, but it is better to give quantitative comparison of these length scales, including polarization vortices.

6. While this study suggests that polarization vortices are dynamic but influenced by chemical heterogeneity, the evidence is not fully conclusive. Additional quantitative analysis (e.g., temperature-dependent elastic vs. inelastic scattering) would strengthen this claim.

7. The connection between local structure results and relaxor theory could be expanded. For example, do vortex structures relate to the frequency dispersion of dielectric response in PMN and PMN-PT system?

Reviewer #4

(Remarks to the Author)

The experimental part of the work is very interesting and exciting; it is well-written (in overall) and perfectly illustrated. I strongly recommend its publication in Nature Communications providing that Authors consider and address properly several minor remarks related to theoretical background.

1) Authors stated in abstract that "results suggest that chemical disorder, via depolarizing and strain fields, stabilizes these polar topologies, offering a route for engineering new dielectric and ferroelectric functionalities". They also mention depolarization field and strain in the Introduction; as well as wrote that "We hypothesize that this swirling topology is such that it simultaneously minimizes strain and depolarizing fields". I believe that these statements are correct, however failed to find/relate corresponding contributions in the theoretical modelling used in the work (namely in the description of the Edwards-Anderson local order parameter) with the strain and depolarization field energy. Moreover, the description of the Edwards-Anderson model is very brief to understand the details, how it was used to the problem. Authors should add more details, basic equations/formulas and parameters used.

2) Since the Authors demonstrate the "pinning of polarization vortices by the local chemistry" (see e.g., figure 5), they should also discuss and/or estimate the role of local flexo-chemical strains. Indeed, the vortex-like polarization is inevitably coupled to the flexoelectricity (see e.g., [<https://doi.org/10.1016/j.actamat.2021.116889>]), which is coupled strongly to the chemical disorder. The mechanism was studied in phenomenological framework for relaxors with ABO₃ structure (see [<https://link.aps.org/doi/10.1103/PhysRevB.98.094102>]). Does the flexo-chemical coupling can be important (or not) for the relaxor PMN-PT system studied in this work?

3) It is demonstrated that "correlations that yield labyrinth patterns with larger correlation lengths and faceting..." (see figure 3). In my opinion, the patterns are not labyrinthic, while they contain some randomly located curled structures. I suggest either using other terminology (like random spots), or to prove the labyrinthine morphology of the patterns. What factors can tune the sizes between the vortex-type features and facets? Polarization gradient energy, local chemistry or/and other factors? How do these structures stable with respect to the temperature change? What is the energy variation between such patterns (I mean two different mazes)?

4) It is not always clear from the figure captions, which panels are experimental data, their treatment, or theoretical calculations. Please improve the captions, where it is relevant.

5) optional. Fonts and labels (a), (b)... in figures are very different in size. Please consider to make the labels a bit smaller.

Version 1:

Reviewer comments:

Reviewer #1

(Remarks to the Author)

Reviewer #2

(Remarks to the Author)

The authors have responded to the reviewers' comments. The manuscript may be accepted in its actual form.

Reviewer #3

(Remarks to the Author)

This revised manuscript investigates emergent topological polarization textures in PMN-PT relaxor ferroelectrics using

advanced structural analysis approach. This work represents a significant contribution to the field and address fundamental questions about the nature of polarization. Authors almost solve all my confusions and questions in the revised version. Therefore, I recommend this manuscript to be published in Nature Communications.

Reviewer #4

(Remarks to the Author)

Authors reply and made changes fully address my remarks and suggestions. Revised work can be recommended for publication.

Response to the reviewers' comments

We would like to thank the reviewers for their positive and constructive comments, which we have addressed as follows:

Reviewer 1:

Comment 1: The author focused on the PMN-30PT and PMN-35PT, can this be observed in other ratio doping or material doping?

Response: While our study focused on PMN-30PT and PMN-35PT, we expect that similar continuous, maze-like polarization textures will persist up to the PT contents where chemical inhomogeneities can still generate significant local strain gradients and internal depolarizing fields. Identifying the cutoff composition will require further studies. Additionally, we believe that similar textures may also develop in other relaxors that exhibit nanoscale chemical disorder. We have included a sentence reflecting this hypothesis at the end of the Discussion (lines 420-421).

Comment 2: It is suggested to add more experimental results to give more convincing conclusion.

Response: Our paper already integrates a wide range of complementary structural probes, including neutron and X-ray total scattering, EXAFS, and 3D reconstructions of diffuse scattering from a single crystal. Additionally, high-quality STEM studies of these materials have been published by others, and our refined configurations align with many of their findings, noting that our models account for both static and dynamic atomic displacements, whereas STEM reveals only the static components. We agree that additional measurements, such as spectroscopies sensitive to lattice dynamics and the chirality of vortices, would be valuable. However, these experiments fall outside the scope of the current study.

Reviewer 2:

Comment 1: . At the end, the authors write about a difference between their picture proposed for PMN-based and other relaxor systems, such as isovalent BaTiO₃-BaZrO₃ solid solutions. Therefore, it is proposed to change the title to "Emergent topological polarization textures in PbMg_{1/3}Nb_{2/3}O₃-PbTiO₃ relaxor ferroelectrics."

Response: We would prefer to keep the current, more general title because we feel that our results apply to multiple relaxor chemistries, even though exceptions inevitably exist. A statement on lines 420-421 addresses this point.

Reviewer 3:

Comment 1: The current refinement procedure employs a sequential approach, first optimizing B-site cation ordering through atom swaps before allowing O atom displacement. The author should justify why they do not employ simultaneous refinement (e.g., using SWAP_MULTI) allowing displacement and swap. An analysis of results obtained from both approaches would

be valuable to demonstrate the rationality of this refinement strategy.

Response: In the Methods section, we provided a detailed justification for using a sequential approach to determining the distribution of Mg and Nb from the diffuse $\frac{1}{2} \frac{1}{2} \frac{1}{2}$ -type superlattice spots. We chose this approach because allowing for cation swapping and atomic shifts invariably drives the heavy Pb and Nb atoms to create a false superlattice – reproducing the $\frac{1}{2} \frac{1}{2} \frac{1}{2}$ reflections – before the true Mg/Nb short-range can emerge. To avoid this artefact, we assume that the diffuse $\frac{1}{2} \frac{1}{2} \frac{1}{2}$ reflections are associated entirely with the ordering of Mg and Nb and the associated oxygen displacements. A close match between our maps of the local chemical order parameter and those obtained using atomic-resolution STEM images [56] testifies to the adequacy of this assumption. When we allowed simultaneous atomic swaps and moves during later stages of the refinements, the result did not change. In the revised version, we have expanded this paragraph in the Methods and added the STEM reference [Ref. 56] to fully clarify our rationale.

Comment 2: Fig.1 presents three distinct order parameters, but the calculated procedures require further clarification, especially the local metric order parameter (Fig.1(a)), and rotational order parameter (Fig.1(d)). Moreover, Figure 1(a) only shows for one layer Pb, and is this same for different layers?

Response: We modified the caption of Fig. 1 (lines 137, 143-147) and the Methods section (lines 752-760) to describe all the order parameters explicitly. We also added a note indicating that the layer in Fig. 1c is the same as in Fig. 1a.

Comment 3: The definition of average angle needs further clarification. Fig.2(b) used the Fourier filtering of the calculated DS amplitude, and any other data included (e.g., total scattering, EXAFS)? If included, what are the weighting factors for these data sets?

Response: We added a figure (Fig. S23) to illustrate the definition of the angle α . With regard to the comment about Fig. 2b, the caption accurately states that it is a result of the Fourier filtering of the calculated DS amplitude. There is no other data involved in such filtering. We added a reference to the Methods section (line 86) describing this procedure to eliminate any ambiguity.

Comment 4: The “Correlations among polar displacements: correlation length vs. PNR size” part (particularly from line 204) requires reorganization to clearly distinguish the contents of Fig.2 and Fig.3.

Response: We agree with the reviewer that the original text was somewhat confusing. We rewrote this paragraph (lines 226-240) and the caption of Fig. 3 (lines 202, 204-205) to clearly distinguish between the contents of Fig. 2 and Fig. 3.

Comment 5: The author pointed that the α -PNR is smaller than the correlation length, but it is better to give quantitative comparison of these length scales, including polarization vortices.

Response: We have added a quantitative estimate (lines 262-263) relating the correlation length for the displacement components to the size of α -PNRs. We feel that the current refinements are insufficient for accurately assessing the sizes of the vortices.

Comment 6: While this study suggests that polarization vortices are dynamic but influenced by chemical heterogeneity, the evidence is not fully conclusive. Additional quantitative analysis (e.g., temperature-dependent elastic vs. inelastic scattering) would strengthen this claim.

Response: We appreciate the suggestion to include the inelastic scattering measurements. As outlined in the paragraph on lines 331-344, our conclusions are primarily based on the ultra-high resolution neutron spin-echo measurements conducted by Stock et al. [32], which definitively indicate the inelastic character of the diffuse scattering in PMN. While we also performed variable-temperature neutron measurements of diffuse scattering using the CORELLI spectrometer at SNS, which allows for the separation of the elastic from the total signal (Methods section and Fig. S8), its 1 meV resolution is insufficient to resolve the low-frequency dynamics in question. Therefore, we believe that Ref. [32] already provides the most conclusive evidence for the dynamic nature of the diffuse scattering, which encodes information about the correlations responsible for the polarization textures reported here. In summary, we feel that no further additions are necessary within the scope of this study.

Comment 7: The connection between local structure results and relaxor theory could be expanded. For example, do vortex structures relate to the frequency dispersion of dielectric response in PMN and PMN-PT system?

Response: We addressed the influence of the polarization textures on the relaxation times in the Discussion section, lines 401-404. However, extending this to claim a more specific link based on the current data would be too speculative.

Reviewer 4:

Comment 1: Authors stated in abstract that “results suggest that chemical disorder, via depolarizing and strain fields, stabilizes these polar topologies, offering a route for engineering new dielectric and ferroelectric functionalities”. They also mention depolarization field and strain in the Introduction; as well as wrote that “We hypothesize that this swirling topology is such that it simultaneously minimizes strain and depolarizing fields”. I believe that these statements are correct, however failed to find/relate corresponding contributions in the theoretical modelling used in the work (namely in the description of the Edwards-Anderson local order parameter) with the strain and depolarization field energy. Moreover, the description of the Edwards-Anderson model is very brief to understand the details, how it was used to the problem. Authors should add more details, basic equations/formulas and parameters used.

Response: We thank the reviewer for asking to clarify the role of theoretical modeling in this work. As noted on lines 90-92, our structural models were derived exclusively from experimental data by refining atomic coordinates against several types of diffraction and spectroscopic measurements without any theoretical input. All “order parameters”, such as local polarization and other metrics, are post-refinement descriptors calculated solely from the refined atomic coordinates. We have removed the reference to the Edwards-Anderson order parameter, as it doesn't apply in our analysis framework. In the revised Methods section (lines 777-780), we provide the formula used to calculate local polarization. Our hypothesis regarding the observed topological textures minimizing strain and depolarizing energies is based on the findings from published computational modeling of topological features in NBT-based ceramics [38] and PZT [39]. We have also modified the text (lines 350-352 and 393-394) to clarify this point.

Comment 2: Since the Authors demonstrate the “pinning of polarization vortices by the local chemistry” (see e.g., figure 5), they should also discuss and/or estimate the role of local flexo-chemical strains. Indeed, the vortex-like polarization is inevitably coupled to the flexoelectricity (see e.g., [<https://doi.org/10.1016/j.actamat.2021.116889>]), which is coupled strongly to the chemical disorder. The mechanism was studied in phenomenological framework for relaxors with ABO₃ structure (see [<https://link.aps.org/doi/10.1103/PhysRevB.98.094102>]). Does the flexo-chemical coupling can be important (or not) for the relaxor PMN-PT system studied in this work?

Response: The reviewer makes an excellent point about the potential importance of the flexoelectric coupling. We agree that the flexoelectric effects can be relevant in PMN-PT, given the observed local strain gradients. We added this discussion to the text (lines 359-361), including the suggested references [Ref. 40, 41].

Comment 3: It is demonstrated that “correlations that yield labyrinth patterns with larger correlation lengths and faceting...” (see figure 3). In my opinion, the patterns are not labyrinthic, while they contain some randomly located curled structures. I suggest either using other terminology (like random spots), or to prove the labyrinthine morphology of the patterns. What factors can tune the sizes between the vortex-type features and facets? Polarization gradient energy, local chemistry or/and other factors? How do these structures stable with respect to the temperature change? What is the energy variation between such patterns (I mean two different mazes)?

Response: We agree that the description of the observed patterns as “labyrinth” was rather vague and have replaced it with “bicontinuous” (line 234). We believe this term provides a better description of the observed morphologies. We added a sentence summarizing the likely factors influencing the sizes of the vortex features (lines 406-408). Quantifying the relative significance of these factors, as well as assessing energies of different mazes, will require dedicated modeling studies.

Comment 4: It is not always clear from the figure captions, which panels are experimental data, their treatment, or theoretical calculations. Please improve the captions, where it is relevant.

Response: We modified the relevant figure captions to indicate whether specific figures reflect fitting results or specific descriptors calculated for the refined configurations.

Comment 5: optional. Fonts and labels (a), (b)... in figures are very different in size. Please consider to make the labels a bit smaller.

Response: We would like to keep the panel labels larger than the axes fonts to make them easily identifiable, even on top of a background. We resized the figures to maintain the consistency of the font sizes used.

Sincerely,

Igor Levin

This paper reconstruct 3D mesoscale polarization maps in the classic relaxor system $\text{PbMg}_{1/3}\text{Nb}_{2/3}\text{O}_3 - \text{PbTiO}_3$, indicating that chemical disorder stabilizes inner polar topologies, offering a route for engineering new dielectric and ferroelectric functionalities. The author addressed the mentioned questions and I suggest that could be accept in this version: